# Sulfur monoxide dimer chemistry as a possible source of polysulfur in the upper atmosphere of Venus

Joseph P. Pinto[1], Jiazheng Li [2✉], Franklin P. Mills [3,4], Emmanuel Marcq [5], Daria Evdokimova [5,6], Denis Belyaev [6] & Yuk L. Yung [2,7]

The abundance of SO dimers $(SO)_2$ in the upper atmosphere of Venus and their implications for the enigmatic ultraviolet absorption has been investigated in several studies over the past few years. However, the photochemistry of sulfur species in the upper atmosphere of Venus is still not well understood and the identity of the missing ultraviolet absorber(s) remains unknown. Here we update an existing photochemical model of Venus' upper atmosphere by including the photochemistry of SO dimers. Although the spectral absorption profile of SO dimers fits the unknown absorber, their abundance is found to be too low for them to contribute significantly to the absorption. It is more likely that their photolysis and/or reaction products could contribute more substantively. Reactions of SO dimers are found to be important sources of $S_2O$, and possibly higher order $S_nO$ species and polysulfur, $S_n$. All of these species absorb in the critical ultraviolet region and are expected to be found in both the aerosol and gas phase. indicating that in-situ high resolution aerosol mass spectrometry might be a useful technique for identifying the ultraviolet absorber on Venus.

[1] University of North Carolina at Chapel Hill, Chapel Hill, NC, USA. [2] Division of Geological and Planetary Science, California Institute of Technology, Pasadena, CA, USA. [3] Australian National University, Canberra, ACT, Australia. [4] Space Science Institute, Boulder, CO, USA. [5] LATMOS/CNRS/Sorbonne Université/ UVSQ, Paris, France. [6] Space Research Institute of the Russian Academy of Sciences (IKI), Moscow, Russia. [7] Jet Propulsion Laboratory, California Institute of Technology, Pasadena, CA, USA. ✉email: jiazheng@caltech.edu

The photochemistry of sulfur species in the upper atmosphere of Venus is not well understood and is poorly quantified. The known parent molecules upwelling from the deep atmosphere are $SO_2$ and OCS. Near and above the cloud tops, these molecules are photolyzed, leading to the production of $S_n$ (n = 1–7), $S_x$ (x = 8+), SO, $SO_3$, and $H_2SO_4$[1]. Significant amounts of sulfur aerosol were predicted in the lower and middle cloud layers by ref. [2] and also ref. [3], which assumes a particle radius of 0.5 μm. Ref. [4] suggested that the dimers of sulfur monoxide (SO), $(SO)_2$, formed from the self-recombination of SO, could contribute substantially to the ultraviolet (UV) absorption found on Venus. In addition, production of $S_nO$ could also occur, as suggested by ref. [5].

There are three main isomeric forms of the SO dimer. Reference[4] concluded that SO recombination yields ~49% as *cis*-OSSO, ~49% as *trans*-OSSO isomers and no more than 2% as the trigonal isomer ($S=SO_2$) which is the lowest energy isomer. Reference [6] considered only the formation of the trigonal form, photolysis of which leads to the formation of S and $SO_2$. Reference[3], based on photochemical modeling, suggested that the abundance of OSSO is too low by two orders of magnitude to explain the near UV absorption of Venus. Photolysis of SO dimer is expected to yield mainly two SO radicals[4], but we will consider other possible reaction paths. Reference [7] found evidence for the formation of the *cis*-OSSO, *trans*-OSSO, and C1-$S_2O_2$ (cyclic OS (=O)S) dimers, along with several other isomeric forms in much lower abundance, following condensation into a solid matrix with $N_2$ and subsequent irradiation. Production of $S_2$ and $SO_2$ was also observed, but the trigonal ($S=SO_2$) dimer form was beneath detection in the observations of ref. [7]. $S_2$ was formed initially from dissociation of the ethylene episulfoxide used as source of SO; further irradiation at 365 nm depleted the 370 nm band, which they ascribed to destruction of syn-OSSO (*cis*-OSSO). This depletion occurred simultaneously with an increase in absorption in the 287 nm band of $S_2$. Formation of $S_2$ requires only 1 eV above that for dissociation of dimer back to SO and could account for the very strong absorption and characteristic vibrational structure seen at 287 nm in their experiments. We suggest that the photochemistry of $(SO)_2$ could provide a significant source of $S_2$ on Venus based on the results of ref. [7], leading ultimately to the production of polysulfur, a candidate for the unidentified UV absorber on Venus[1] and possibly also to production of polysulfur oxides[5].

In this work, we use a photochemical model to assess the contributions of the proposed dimer chemistry to the production of polysulfur. We find that the SO dimers are more likely to be important intermediaries in the formation of more complex S species that could be responsible for the UV absorption. Our model chemistry is also applied to interpret data from the European Space Agency's Venus Express (VEx) mission that was recently processed and analyzed and to provide input to plans for future missions.

## Results

Figure 1 presents the mixing ratios of $SO_2$, OCS, and SO in the middle atmosphere using an $SO_2$ mixing ratio of 3.0 ppm, which we have adopted for our standard model at the lower boundary (58 km), based on ref. [8] and 0.3 ppm for OCS, based on ref. [9]. The profiles of the rates of the reactions in Supplementary Table 1 are shown in Figure Supplementary Fig. 1. Comparison of the model with several observational datasets at higher altitudes is also shown in Fig. 1. As can be seen (curve a), the model is in reasonable agreement with $SO_2$ mixing ratios at 70 km during the first four years (2006 through 2009) of VEx measurements[10]. Calculated values are slightly higher than the interquartile range and the mean of VEx measurements within ±20° latitude; but the data include a large number of spikes, as indicated by the difference between the median and mean observations. As an indication of the model sensitivity to the choice of $SO_2$ at the lower boundary, we also show the model-measurement comparison using an $SO_2$ mixing ratio of 0.3 ppm at the lower boundary in Supplementary Fig. 2. This lower boundary condition could also be seen as reflecting in some ways the lower cloud top $SO_2$ mixing ratios seen from the start of 2010 through 2014. Calculated values are within the interquartile range and are quite close to the mean of VEx measurements within ±20° latitude. As is the case for the earlier period in the record, the mean differs significantly from the median because of the existence of a large number of concentration spikes, which are roughly three orders of magnitude greater than median levels.

Modeled $SO_2$ (thin solid blue line) is also consistent with Hubble Space Telescope (HST) data (dashes b) of ref. [11] and with the upper limit derived from ground-based submillimeter observations (dashes d) of ref. [12] from 85 to 100 km and is in reasonable agreement with Spectroscopy for Investigation of Characteristics of the Atmosphere of Venus/Solar Occultation at Infrared (SPICAV/SOIR) measurements from 90 to 100 km (curve c) of ref. [13] collected from 2006 through 2014. Note that the upper level $SO_2$ mixing ratios (curve c) obtained by ref. [13] from 2006 through 2009 tend to be higher than those from 2010 through 2014 in a manner similar to $SO_2$ observed at 70 km by ref. [10]. Both ground-based submillimeter spectroscopy[12] and solar and stellar occultations[13] with one standard deviation (curve c and e) show higher $SO_2$ mixing ratio at high altitudes (z > ~85 km) compared to lower altitudes. There are two possible sources of $SO_2$ to be considered at these altitudes, meteorite ablation and photolysis of $H_2SO_4$. We have included ablation of meteoritic material, which is a very minor source of S to the upper atmosphere of Venus based on current estimates of meteoric input[14] and S content[15]. The calculated profile of $H_2SO_4$ is within limits obtained by ref. [16]. Photolysis of $H_2SO_4$ is also included based on calculations of its absorption spectrum by ref. [17], which provided evidence for the existence of a long-wavelength tail that

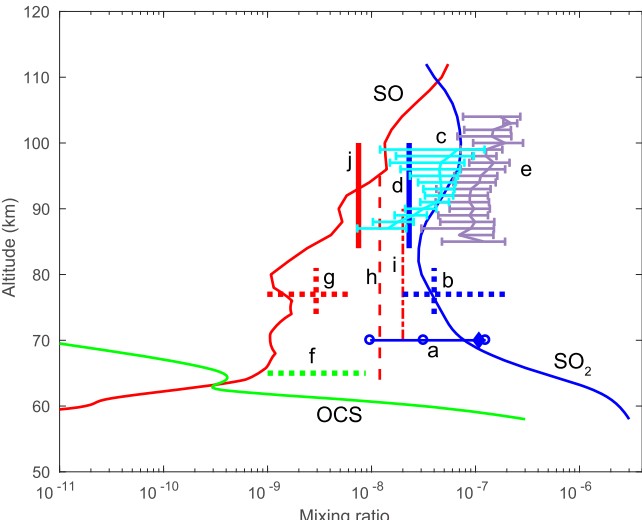

**Fig. 1 Modeled profiles of SO2, OCS, and SO.** Comparison of modeled profiles of $SO_2$, OCS and SO with observations for $SO_2$ = 3.0 ppm and OCS = 0.3 ppm at 58 km, the lower boundary of the model. Model profiles are shown as thin solid lines: (blue) $SO_2$, (green) OCS, (red) SO. Data sources: **a** $SO_2$, interquartile range from 2006 through 2009, diamond shows mean of distribution, ref. [10]; **b** $SO_2$, ref. [11]; **c** $SO_2$, ref. [34] solar occultation with 1-σ error bars; **d** $SO_2$, ref. [12]; **e** $SO_2$, ref. [34] stellar occultation with 1-σ error bars; **f** OCS, ref. [35]; **g** SO, ref. [11]; **h** SO, ref. [14]; **i** SO, ref. [13]; **j** SO, ref. [12].

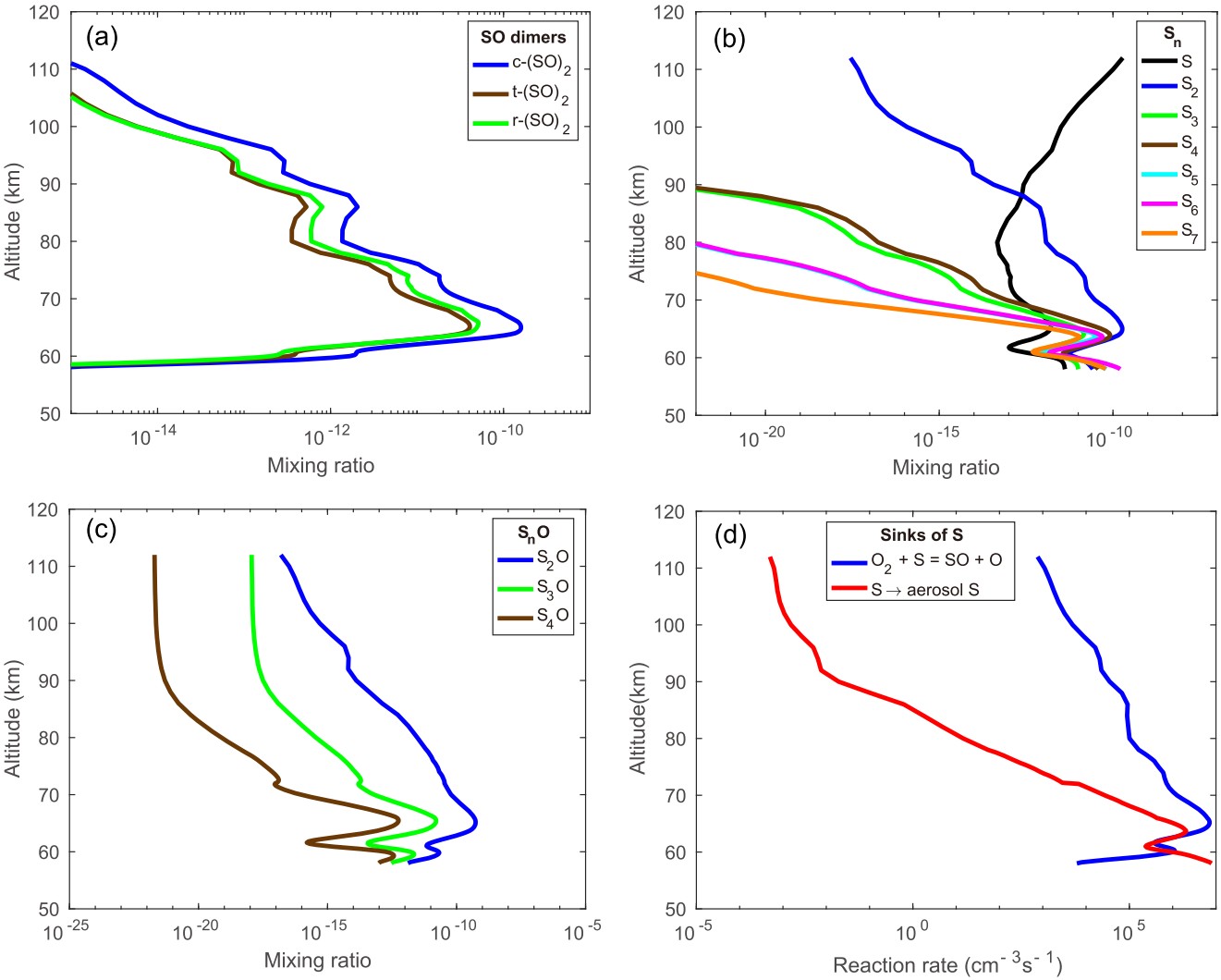

**Fig. 2 Model results. a** Modeled profiles of *cis*-(SO)₂, *trans*-(SO)₂, and trigonal-(SO)₂. **b** Modeled profiles of $S_n$ species. **c** Modeled profiles of $S_2O$, $S_3O$ and $S_4O$. **d** Comparison of production of aerosol S versus recycling of S back to SO.

substantially increases the photolysis rate of $H_2SO_4$ in the current model.

The reasons for the differences between observed and calculated $SO_2$ and OCS for some datasets shown in Fig. 1 are not entirely clear, but some of the applicable datasets share the common feature that they were obtained on relatively short time scales. The $SO_2$ record at 70 km exhibits dramatic spatial and temporal variability with $SO_2$ mixing ratios in the equatorial region spanning three orders of magnitude from ppb to ppm levels. Data for some datasets could have been obtained during the brief periods when vertical transport was greatly enhanced compared to the mean values used here. Examining variability in transport as a source of variability in mixing ratios of $SO_2$ and OCS is beyond the scope of this paper and is best accomplished using a multidimensional model incorporating radiative, chemical and dynamical feedbacks.

Reasonably good agreement of modeled SO with Hubble Space Telescope observations of ref. [11] and the mean SO abundance determined by submillimeter observations of ref. [12] at higher altitudes is found using a model with 3.0 ppm $SO_2$ at the lower boundary. Larger model-observation differences are found for the SO mixing ratios observed by ref. [18] and ref. [19] beneath an altitude of about 80 km. Reference [18] concluded that IUE data were best fit with an SO mixing ratio of $20 \pm 10$ ppb above ~70 km with

no SO beneath that altitude; ref. [19] derived a best fit to their data with a constant mixing ratio of SO of $12 \pm 5$ ppb for $z \geq 64$ km and falling off sharply beneath 64 km. As can be seen from Fig. 1, our calculated mixing ratio of SO is 0.14 ppb at 64 km, increasing with height. As can also be seen from Fig. 1, our calculated mixing ratio of SO is ~1 ppb at 70 km.

The $SO_2$ profiles shown in Supplementary Fig. 2 (solid blue line) calculated using 0.3 ppm at the lower boundary consistently underpredict the higher altitude observations. As seen in Supplementary Fig. 2, a mixing ratio of 0.3 ppm for $SO_2$ applied at the lower boundary leads to an SO mixing ratio that is slightly too low compared to observations (dashes g, h, i, j). However, much of the disagreement may simply be due to the sparse nature of observations of SO that are not able to capture the extent of spatial and temporal variability as was possible for $SO_2$.

The mixing ratios of the three isomers of (SO)₂, c-(SO)₂, t-(SO)₂, and r-(SO)₂, are shown in Fig. 2a. Our values are consistent with the model of ref. [3] but are far lower than those calculated by ref. [4], particularly in the crucial layer beneath 70 km. The reason is that ref. [4] fixed the mixing ratio of SO to 12 ppb at 64 km based on the model of ref. [19] while decreasing it to 3 ppb at 70 ppb based on the microwave observations of ref. [12] and then increasing it to 150 ppb at 96 km based on ref. [20]. The altitude profile of SO derived by ref. [19] should be reevaluated in the

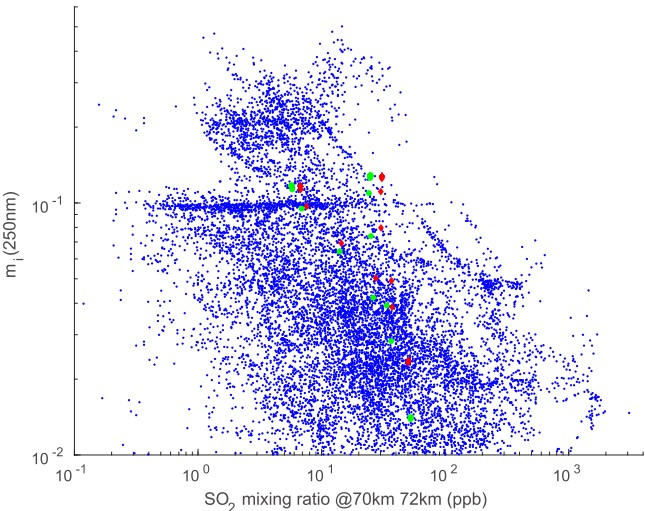

**Fig. 3 Correlation between $S_x$ and $SO_2$.** Scatterplot showing imaginary index of refraction at 250 nm versus $SO_2$ at 70 km from Venus Express[10] (blue dots) compared to the scatterplot of $S_x$ versus $SO_2$ at 70 km (red diamonds) and 72 km (green dots) obtained from our model. The mixing ratios of $S_x$ in our model are multiplied by $2 \times 10^5$ in order to compare with the imaginary index of refraction derived from ref. [10].

context of the shape of the modeled profile, which shows an increase in the mixing ratio of SO with height. This is a more likely shape for the SO mixing ratio profile based on measurements of ref. [11], ref. [12] and current understanding of photochemistry. It must also be kept in mind that conditions on Venus are highly spatially and temporally variable, so short-term measurements might not represent conditions at other times. The mixing ratios of $S_n$ ($n = 1$–7) are shown in Fig. 2b and $S_nO$ ($n = 2$–4) are shown in Fig. 2c.

There is competition between production of reduced aerosol and recycling of oxidized S. It is clear from Fig. 2d, that the reaction, $S + O_2 \rightarrow SO + O$, is the primary sink for S throughout the modeling domain, except in a few thin layers at about 64 km and below. This demonstrates the difficulty of producing $S_n$ via S atoms, which are rapidly converted to SO in the presence of $O_2$. In our current model, production of $S_2$ does not directly involve S atoms, as it is derived from alternative paths involving SO dimer photochemistry and as seen below from catalytic cycles involving Cl, thus bypassing the rapid recycling of S to SO by $O_2$. Sensitivity studies carried out by varying the rate coefficients in Supplementary Table 1 uniformly upward and downward by a factor of ten indicate that species concentrations are typically within a factor of two of their values shown in Supplementary Table 1.

Cl has been shown to facilitate the stability of CO[21–23] Cl also exerts strong influence on the production of $S_x$. The column production rate of aerosol (on an S basis) is $8.3 \times 10^{11}$ cm$^{-2}$ s$^{-1}$ in our standard model ($SO_2 = 3.0$ ppm, HCl = 0.4 ppm) but it is reduced to $6.4 \times 10^{11}$ cm$^{-2}$ s$^{-1}$ for HCl = 0.2 ppm. At the lower $SO_2$ mixing ratio ($SO_2 = 0.3$ ppm) used at 58 km, the results are even more dramatic. Column production rates of aerosol drop from $2.9 \times 10^{11}$ cm$^{-2}$ s$^{-1}$ to $1.1 \times 10^{11}$ cm$^{-2}$ s$^{-1}$ for a decrease in HCl mixing ratio from 0.4 to 0.2 ppm. Although changes in major species such as $SO_2$ associated with changes in HCl are relatively minor, order of magnitude changes in S and Cl-S species are seen. This strong sensitivity results in part from Cl reactions depleting $O_2$, the major sink for S, through recombination of $CO_2$. This is similar to what was found for production of $S_2$ via chlorosulfanes[24].

There are a number of intriguing clues linking $SO_2$ photochemistry to the unknown absorber. As shown in Fig. 2d, the maximum production rate of absorbing aerosol in our model

occurs in the lower half of the upper cloud deck, but noticeable depletion of $SO_2$ occurs only at higher altitudes. Our results are consistent with VEx observations showing an inverse relation between $SO_2$ at 70 km and UV absorption at 250 nm[10], as shown in Fig. 3. The data points in Fig. 3 are obtained from the time steps towards equilibrium when running our model and are meant to provide an indication of the adequacy of the model's production rate of absorbing aerosol. The density of $S_x$ is found to be negatively correlated with the local $SO_2$ mixing ratio, which is consistent with the observations of ref. [10]. However, as noted by ref. [25], the lifetime of the UV absorber is much longer than that of $SO_2$, and as a result, the ratio of their abundances is strongly affected by transient atmospheric dynamics, in particular convective activity. Upward transport in the ascending branch of the Hadley Cell could bring absorber upward from where it is formed in the lower half of the upper cloud deck. Events such as these would reduce the strength of any correlation between $SO_2$ and $S_x$. Since our one-dimensional model is meant to simulate mean conditions at low latitudes, it cannot encompass the full range of conditions sampled by *Venus Express*. It therefore cannot simulate the behavior of $SO_2$ and the UV absorber undergoing poleward transport; a detailed simulation of their latitudinal behavior requires a two- or three-dimensional model.

## Discussion

The SO dimers are more likely to be important intermediaries in the formation of more complex S species that could be responsible for the UV absorption, rather than being the UV absorbers themselves. Cl is an important modulator of important chemical processes maintaining the stability of $CO_2$ and the production of condensable sulfur species. These species include $S_x$ and $S_n$, which have been the subject of several studies (see e.g. ref. [24] and references therein), and possibly $S_nO$[5]. In our current model, which only includes condensational loss to existing particles and does not consider new particle formation, $S_2O$ is the major condensable species followed by $S_2$ and $S_4$. Although it's still not clear exactly what is driving the overall temporal behavior of the albedo at 365 nm, a possible explanation is that it is driven by the variability in $SO_2$ levels, in particular through the formation of some of the UV absorbers discussed in the literature (e.g. ref. [26]) and/or from polysulfur and polysulfur oxide species. The implications of our model are that more complex sulfur compounds, beyond OCS, SO, $SO_2$, $SO_3$, and $H_2SO_4$, that are also produced by $SO_2$ photochemistry could contribute to the absorption of UV in the enigmatic 320–400 nm range. Gas phase abundances of polysulfur and $S_nO$ by themselves are not capable of accounting for the NUV absorption but a combination of their condensed and gas phase forms (or in addition to $FeCl_3$) might.

Our base modeling provides reasonable fits to available data, although the significance of these results is not entirely clear given the sparsity of data. The sporadic nature of observations of key species precludes making definitive conclusions regarding the major mechanisms involved in production of the enigmatic UV absorber and for understanding the photochemistry occurring in the upper atmosphere of Venus, especially given the large spatial and temporal variability of species that has been observed, e.g. by *Venus Express*. Rather, a more concentrated approach measuring a number of interacting species simultaneously would provide much needed input and hence greater confidence in our understanding of major chemical processes. In situ measurements using high resolution aerosol mass spectrometry (e.g. ref. [27]) might be a way to identify the UV absorber on a future probe. Such measurements would be able to supply information not only identifying the major elements (S, O, C, Cl, N, and H) that may be present in the aerosol, but also their isotopic composition. Long

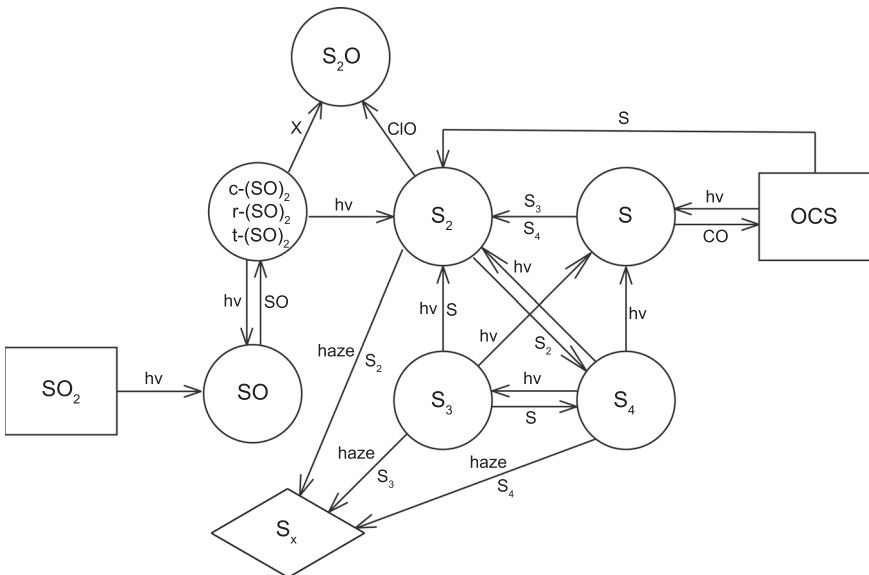

**Fig. 4 Major reactions in the model.** Schematic diagram showing the main routes for producing $S_x$ (shown in the diamond) from parent molecules, $SO_2$ and OCS (shown in rectangles) in the model. Intermediate species are shown in the circles. Species needed for the reactions are written next to the arrows. X can be O, H, NO, S, SO, $S_2$.

term observations, such as those provided by *Venus Express* are crucial for integrating radiative, chemical and dynamical processes occurring on Venus.

Major uncertainties in the chemistry involve the reaction kinetics of SO and its dimers. Laboratory data on which the formation of SO dimers are calculated from ref. [28] and need to be updated. The photochemistry of the dimers and subsequent reactions of products also need to be examined in the laboratory. Likewise, kinetic data for many of the key reactions involving various $S_n$ and $S_nO$ species are either lacking or need to be updated. In other words, the key uncertainties in both the laboratory and Venus data essentially remain the same as given by ref. [8]. As noted by ref. [29], the terrestrial stratospheric sulphate layer (Junge layer), which is an important regulator of the Earth's climate and the abundance of ozone, is highly similar to the upper haze layer on Venus. So, we think the results described here could be relevant for issues in stratospheric aerosol chemistry, namely, the evolution of stratospheric volcanic eruption clouds and geoengineering of the Earth's climate.

## Methods

Our photochemical model of Venus is based on the Caltech/JPL KINETICS one-dimensional (1-D) model[8,9,21,24,29]. In brief, it is a 1-D diurnal-average steady-state model at the equator, extending in the vertical from 58 to 112 km with 0.2 km spacing from 58 to 78 km and 2 km spacing from 78 to 112 km. Vertical transport is parameterized via eddy diffusion. Radiative transfer processes in the model are the same as were described in ref. [8]. Continuity equations are solved for 60 variable species. Mixing ratios of several key species are fixed throughout: $CO_2 = 0.965$; $N_2 = 0.035$; $H_2O = 1$ ppm. The model contains ~500 thermal and photochemical reactions. Important chemical processes, including a high-altitude source of $SO_2$ from $H_2SO_4$ photolysis are based mainly on ref. [8]. The unusual high nighttime temperature at 95 km in ref. [8] is revised in this model. Constraint on $H_2SO_4$ is also applied in the model to fit the existing observations[16]. Photoabsorption cross sections and reaction rate coefficients are taken mainly from NASA and IUPAC compilations and lab studies[30–33]. The rate coefficient of the reaction $S + CO + M \rightarrow OCS + M$, $3.0 \times 10^{-33} \exp(-1000/T)$ cm$^6$s$^{-1}$, is adopted from ref. [34]. Lower boundary conditions are set for $CO = 45$ ppm, $HCl = 0.4$ ppm, $SO_2 = 3.0$ ppm, $OCS = 0.3$ ppm, and $NO = 5.5$ ppb in our standard model. All other species produced within the model domain are allowed to flow downward with the eddy velocity (K/H). At the model's upper boundary, a zero net flux condition is applied for CO, HCl and NO with respect to $CO_2$, $Cl + H$, and $N + O$; for all other species a zero-flux condition is applied.

The sulfur cycle in our model is summarized in Fig. 4, based on an update of Figure A1 of ref. [9]. The new reactions are summarized in Supplementary Table 1. The most important reactions in our chemical scheme are the formation of SO

dimers, $(SO)_2$, followed by their photolysis,

$$SO + SO + M \rightarrow c-(SO)_2 + M \tag{1}$$

$$\rightarrow t-(SO)_2 + M \tag{2}$$

$$\rightarrow cyclic\ (SO)_2 + M \rightarrow r-(SO)_2 \tag{3}$$

$$c-(SO)_2 + hv \rightarrow SO + SO \tag{4}$$

$$\rightarrow S_2 + O_2 \tag{5}$$

$$t-(SO)_2 + hv \rightarrow SO + SO \tag{6}$$

$$r-(SO)_2 + hv \rightarrow S + SO_2 \tag{7}$$

where M is a third body, $c-(SO)_2$, $t-(SO)_2$ and $r-(SO)_2$ are, respectively, the *cis-*, *trans-*, and trigonal- isomers of the SO dimer[4]. There is a fundamental difference between the formation of $S_2$ by the above reactions and the previous mechanism[9],

$$OCS + hv \rightarrow S + CO \tag{8}$$

$$OCS + S \rightarrow S_2 + CO \tag{9}$$

because this path for forming $S_2$ may be aborted by the competing reaction,

$$S + O_2 \rightarrow SO + O \tag{10}$$

thereby reducing the yield of $S_2$ formation from OCS. By contrast, the pathways for $S_2$ formation via $c-(SO)_2$ and possibly $r-(SO)_2$ do not involve the production of S atoms, and hence could not be short-circuited by $O_2$.

Once $S_2$ is produced, it readily self-propagates to form higher polymers of S via reactions such as:

$$S_2 + S_2 + M \rightarrow S_4 + M \tag{11}$$

$$S_4 + S_2 + M \rightarrow S_6 + M \tag{12}$$

$$S_4 + S_4 + M \rightarrow S_8 + M \tag{13}$$

$S_n$, where $n = 8$ or larger, is lumped together as $S_x$.

A number of abstraction reactions involving $(SO)_2$

$$(SO)_2 + X \rightarrow S_2O + XO\ \ X = O, H, NO, S, SO, S_2 \tag{14}$$

producing $S_2O$ are also included. In addition, reactions involving Cl, such as

$$ClO + S_2 \rightarrow S_2O + Cl \tag{15}$$

and

$$ClS + SO \rightarrow S_2O + Cl \tag{16}$$

are also important sources of $S_2O$. Once formed $S_2O$ can either photolyze or start

polymerization via reactions, such as

$$S_2O + S + M \rightarrow S_3O + M \qquad (17)$$

perhaps producing higher order $S_nO$, in a manner similar to that for producing $S_n$.

## Data availability
The model data and observational data that support the findings of this study are available from the corresponding author on reasonable request.

## Code availability
The code for the photochemical model written in Fortran and the post processing of the data written in Matlab is available from the corresponding author on reasonable request.

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

## Acknowledgements
This research was supported by NASA Grant P1969079 under subcontract to the Space Science Institute (SSI) and NASA grant NNX16AN03G to SSI. D.E. and D.B. (IKI) acknowledge funding from the Russian government (topic VENERA). D.E. acknowledges the support by the French Government Scholarship Vernadski for PhD students.

## Author contributions
J.P.P., J.L., F.P.M., and Y.L.Y. contributed to the paper writing. J.P.P., J.L., and F.P.M. carried out the modelling work. E.M., D.E., and D.B. provided the data from Venus Express. Y.L.Y. supervised the research.

## Competing interests
The authors declare no competing interests.
