## [Peer Review File · Nature Communications]

REVIEWER COMMENTS

Reviewer #1 (Remarks to the Author):

The manuscript models sulfur species in the Venus upper atmosphere with the Caltech/JPL kinetics 1D model, with updated data compared to previous modelling and including the more recently proposed (SO)₂ chemistry. The model is a nice update with new conclusions that the recently proposed OSSO concentration are likely to high, but that SO dimers are an important source of S₂O and possibly poly sulphur.

The manuscript is clearly written and easy to follow.

It seems that the results are very dependent on the SO concentration, which does not seem to be that well constrained. Also the rate constants included in Table S1 for many of the reactions are the same as are the J-values. In Frandsen 2016, they report calculated oscillator strengths for c-(SO)₂ and r-(SO)₂ that are quite different, thus I would expect quite different J values. It would be nice to see some kind of sensitivity of the simulation to the parameters in Table S1.

The difference between Figure 2a and 2b, is very minor, perhaps better to just some one of them, and just state it.

Minor

Make it clear that (SO)₂ means OSSO, not SOSO, which is also mentioned.

Figure 1, Should t-(SO)₂ also be included in square box?

Page 7, 2010-20014, typo

Reviewer #2 (Remarks to the Author):

This paper is an update of the photochemical model of Venus' atmosphere at 58-112 km by Zhang et al. (2012) using the recent progress in the (SO)₂ chemistry. The authors claim this chemistry as a source of polysulfur on Venus. My major comments are below.

1. Clear observational evidences both pro and contra polysulfur and sulfur aerosol are lacking. Polysulfur and sulfur aerosol may contribute to the NUV absorption on Venus. However, they cannot be the major absorber because of a poor spectral fit to the NUV absorption. The calculated abundances of SX and SXO are too low to explain the NUV absorption.

2. The major assumption of the model is that the S₂ + O₂ branch is equal to that of SO + SO in the cis-OSSO photolysis. However, Frandsen et al. (2016) consider SO + SO as the only branch of photolysis of OSSO. Wu et al. (2018) observe a peak at 287 nm and assign it to S₂ but do not refer it to the S₂ + O₂ photolysis branch. Actually the 287 nm peak is the strongest before the irradiation; therefore S₂ is not a product of photolysis. Evidently photolysis of OSSO breaks the weakest bond OS-SO, which is $\approx 1 \text{ eV} = 1.24 \text{ mu}$ with peak absorptions at 320 and 365 nm (Frandsen, Fig. S7). Formation of S₂ + O₂ requires $\approx 2 \text{ eV}$ and very significant rearrangement of the bonds that is low probable.

I tried to find an analog of OSSO in the JPL photochemical compilation that includes in detail various photolysis branches of species. The closest analogs are H₂O₂, Cl₂O₂, and F₂O₂. Two branches that are analogs of SO + SO and O + S₂O exist only for each species. Therefore the adopted S₂ + O₂ yield looks reasonable at, say, $\approx 1\%$, not 50%.

3. The model disagrees with the observations of OCS, while the models by Krasnopolsky (2012, 2018) agree.

Krasnopolsky (2007, *Icarus* 191, 25-37) argued that the spin considerations are weaker for heavy species. Therefore the spin-forbidden reaction S + CO + M is faster than O + CO + M, and the adopted rate coefficient is $3e^{-33}/\exp(1000/T)$, much greater than $4e^{-33}/\exp(1940/T)$ in your model. This reaction balances photolysis of OCS at 47-58 km using S from photolysis of OCS and above 70 km using S from SO + SO \rightarrow SO₂ + S. OCS decreases steeply above 58 km in Krasnopolsky (2018), but not so steeply as in your model. The abundance at 65 km is very sensitive to small variations of eddy diffusion, which is smaller in Krasnopolsky (2018) than in your model by a factor of 5. The large 2 km steps in your model above 60 km may be insufficient for fine tuning of the photochemistry at 60-75 km as well. The text should be properly changed to account for these considerations.

4. The previous concerns to Zhang et al. (2012) remain: (i) the nighttime T $\approx 250 \text{ K}$ at 95 km contradicts to temperatures of the O₂ nightglow observed by four independent teams with a mean value of 187 K at 95 km, to empirical models (VIRA, Hedin et al. 1983) that are based on the accelerometers at the entry probes, radio occultations, and sounding at 15 mu, and to VTGCM; (ii)

the abundance of H₂SO₄ vapor exceeds the upper limit observed by Sandor et al. (2012) and exceeds by orders of magnitude that calculated by Krasnopolsky (2011); (iii) column photolysis rate of H₂SO₄ is close to total delivery of aerosol above 90 km.

Some other comments:

44: Wu et al. (2018) consider six and mention more than ten isomers.

49, change to: ... (Frandsen et al. 2016), and Krasnopolsky (2018) updated his model using these data. However, the predicted abundance of OSSO is smaller than that required to explain the NUV absorption of Venus by two orders of magnitude. Here we consider other possibilities of the S₂O₂ photolysis.

53-54: the S=SO₂ dimer does not have a peak in Fig. 4 in Wu et al. This does not mean that this dimer is lacking.

61: sulfur aerosol absorbs in NUV but poorly fits to the observed NUV absorption.

Significant abundance of sulfur aerosol is predicted in the lower cloud layer (Krasnopolsky 2016, Icarus 274, 33-36). That paper is relevant to your problem and should be mentioned.

117: S + CO + M is much more effective than S + OCS in Krasnopolsky (2012, 2018).

124: r-(SO)₂ involves the production of S atoms.

Actually S₂ is removed in part by photolysis and in the reaction with O.

138-153: quantitative assessments of the reaction rates are lacking, and it is not clear, which reactions are either essential or negligible.

159: according to the Venera 14 observations at 320-390 nm, SO₂ = 10 ppm at 57 km (Krasnopolsky 1986, Photochemistry of the Atmospheres of Mars and Venus, Springer, p.152).

179-180, Figure 2: maximum values of SO₂ and SO from Sandor et al. (2010) are shown. Their mean abundances are smaller by a factor of 4, more representative, and agree with the ALMA observation (Encrenaz et al. 2015). This overestimation is used and should be corrected throughout the paper.

190-192: fitting the OCS data observed near 65 km using 100 ppm at 58 km means that the model disagrees with the observations. My model for the lower atmosphere (Krasnopolsky 2013, Icarus 225, 570-580) predicts OCS = 140 ppb at 47 km, and this value is the OCS boundary condition at 47 km in Krasnopolsky (2018) that agrees with the OCS observations.

301, change to: ...that may be responsible....

300-314: the calculated abundances of SX and SXO are rather low to explain the NUV absorption. The alternative explanation by iron chloride (Krasnopolsky 2017, Icarus 286, 134-137) should be mentioned here.

316-330: I do not share the opinion that “the sporadic nature of observations of key species precludes making definitive conclusions....”

The authors are the well-known authorities in Venus’ photochemistry. Maybe, some differences between our opinions on some aspects of this photochemistry are helpful for our science.

Vladimir Krasnopolsky

REVIEWER COMMENTS

Reviewer #1 (Remarks to the Author):

We thank the reviewers for taking the time to read our manuscript.

The manuscript models sulfur species in the Venus upper atmosphere with the Caltech/JPL kinetics 1D model, with updated data compared to previous modelling and including the more recently proposed (SO)₂ chemistry. The model is a nice update with new conclusions that the recently proposed OSSO concentration are likely to high, but that SO dimers are an important source of S₂O and possibly poly sulphur.

The manuscript is clearly written and easy to follow.

It seems that the results are very dependent on the SO concentration, which does not seem to be that well constrained. Also, the rate constants included in Table S1 for many of the reactions are the same as are the J-values. In Frandsen 2016, they report calculated oscillator strengths for c-(SO)₂ and r-(SO)₂ that are quite different, thus I would expect quite different J values. It would be nice to see some kind of sensitivity of the simulation to the parameters in Table S1.

Measurements of many key rate coefficients are not available, so choices had to be made based on data for analogous reactions as used in earlier studies. For the new rate coefficients of thermal reactions (k₈-k₂₃) listed in the SI, we adjusted them either downward or upward by a factor of 10 to provide an indication of their potential importance with respect to their overall effect on the dimers and on the products, S_n and S_nO. The results of sensitivity tests for those reactions are shown below. In each figure, the solid lines represent the original mixing ratio profiles of the chemical species, while the dashed lines and the dotted lines represent the profiles with rate coefficients adjusted upward and downward by a factor of 10, respectively. The sensitivity test results show that the simulated species concentrations are typically within a factor of two of their values as shown in the figures below.

The overall effects are summarized in the text in the results and conclusion sections. Please see lines 259-261 in the main text.

We have adjusted the J-values for trigonal (SO)₂ based on sums of oscillator strengths for wavelengths > 200 nm (which is the portion of the solar spectrum that penetrates below 70 km) given in Tables (S7-S9) of the SI of Frandsen et al. (2016).

The difference between Figure 2a and 2b, is very minor, perhaps better to just show one of them, and just state it.

We have followed the referee's recommendation by removing the sensitivity case with SO₂ = 0.3 ppm from the body of the text and inserted that figure into the SI as Figure S1.

Minor

Make it clear that (SO)₂ means OSSO, not SOSO, which is also mentioned.

Done

Figure 1, Should t-(SO)₂ also be included in square box?

Done

Page 7, 2010-20014, typo

Done

Reviewer #2 (Remarks to the Author):

This paper is an update of the photochemical model of Venus' atmosphere at 58-112 km by Zhang et al. (2012) using the recent progress in the (SO)₂ chemistry. The authors claim this chemistry as a source of polysulfur on Venus. My major comments are below.

1. Clear observational evidences both pro and contra polysulfur and sulfur aerosol are lacking. Polysulfur and sulfur aerosol may contribute to the NUV absorption on Venus. However, they cannot be the major absorber because of a poor spectral fit to the NUV absorption. The calculated abundances of SX and SXO are too low to explain the NUV absorption.

This lack of evidence is noted in the conclusions along with a possible way to remedy this shortcoming. Sulfur and sulfur oxide aerosol along with non-sulfur containing species, e.g., FeCl₃, could contribute to the near UV absorption. With regard to the goodness of fit of candidate absorbers, (see Figure 15 in Perez-Hoyos et al. (2018)), it can be seen that it is very difficult to assign a single species as the dominant absorber, but S₂O as noted by Perez-Hoyos et al. and S_xO as noted by Hapke and Graham (1989) comes close and as we note in the text, it is the major condensable species in our model.

Figure 15. A comparison of the relative absorption (in arbitrary units) of some of the candidates for the UV absorber proposed so far with the model results obtained in this work. The gray area is used for our model results, with the black dashed line being used for best fitting values, and maximum and minimum absorption values being indicated with solid black lines.

The reviewer is correct in saying that gas phase abundances of S_x and SnO by themselves are not capable of accounting for the NUV absorption but a combination of condensed and gas phases (or a combination with $FeCl_3$) might. We are not claiming that we have solved the mystery of the UV absorber, as we do not yet have a fully interactive gas phase chemical-aerosol microphysical module. So, we cannot explicitly calculate the abundance of particles produced from gas phase chemistry and hence the composite NUV absorption profile. We can only note that production rates of S bearing particles are substantial, $3.2 \times 10^{11} \text{ cm}^{-2} \text{ s}^{-1}$, in our base model. Combined with a downward eddy velocity at the lower boundary we get an abundance of $\sim 5 \times 10^{17} \text{ cm}^{-2}$ as an upper limit. Please see lines 39-40, 67-68 and 313-315.

2. The major assumption of the model is that the $S_2 + O_2$ branch is equal to that of $SO + SO$ in the cis-OSSO photolysis. However, Frandsen et al. (2016) consider $SO + SO$ as the only branch of photolysis of OSSO. Wu et al. (2018) observe a peak at 287 nm and assign it to S_2 but do not refer it to the $S_2 + O_2$ photolysis branch. Actually the 287 nm peak is the strongest before the irradiation; therefore S_2 is not a product of photolysis. Evidently photolysis of OSSO breaks the weakest bond OS-SO, which is $\approx 1 \text{ eV} = 1.24 \text{ μm}$ with peak absorptions at 320 and 365 nm (Frandsen, Fig. S7). Formation of $S_2 + O_2$ requires $\approx 2 \text{ eV}$ and very significant rearrangement of the bonds that is low probable.

I tried to find an analog of OSSO in the JPL photochemical compilation that includes in detail various photolysis branches of species. The closest analogs are H_2O_2 , Cl_2O_2 , and F_2O_2 . Two branches that are analogs of $SO + SO$ and $O + S_2O$ exist only for each species. Therefore the adopted $S_2 + O_2$ yield looks reasonable at, say, $\approx 1\%$, not 50%.

We thank the referee for pointing out this typo. We didn't assign equal weights to the production of $S_2 + O_2$ and $SO + SO$ in our model. Instead, we chose a value of 0.1 for the photolysis of cis-OSSO. Cis-OSSO is a planar trapezoid. When excited in bending mode it can become more horseshoe shaped and there is the possibility of forming an O-O bond. As hinted at by the reviewer, this process has been acknowledged in the JPL review and was initially suggested for formation of Cl_2 and O_2 from ClO dimer photolysis in the Antarctic stratosphere. This process however requires more energy than at the

dissociation limit. However, solar photons with energies much larger than the difference between an S-S bond and an S=O bond are available. So, substantial bond rearrangement is not involved.

We have acknowledged that S₂ in Wu's experiment initially came from dissociation of the ethylene episulfoxide used as source of SO. Closer inspection of Figure 4 of Wu et al. reveals that the S₂ peak at 287 nm decreases after initial irradiation, further irradiation at 365 nm depletes the 370 nm band which they ascribe to destruction of syn-OSSO (cis-OSSO). This occurs simultaneously with a (small) increase in the 287 band (S₂). So, there is experimental support for a weak channel producing S₂ from photolysis of OSSO. Please see lines 45-62.

3. The model disagrees with the observations of OCS, while the models by Krasnopolsky (2012, 2018) agree.

Krasnopolsky (2007, Icarus 191, 25-37) argued that the spin considerations are weaker for heavy species. Therefore, the spin-forbidden reaction S + CO + M is faster than O + CO + M, and the adopted rate coefficient is $3e-33/\exp(1000/T)$, much greater than $4e-33/\exp(1940/T)$ in your model. This reaction balances photolysis of OCS at 47-58 km using S from photolysis of OCS and above 70 km using S from SO + SO → SO₂ + S. OCS decreases steeply above 58 km in Krasnopolsky (2018), but not so steeply as in your model. The abundance at 65 km is very sensitive to small variations of eddy diffusion, which is smaller in Krasnopolsky (2018) than in your model by a factor of 5. The large 2 km steps in your model above 60 km may be insufficient for fine tuning of the photochemistry at 60-75 km as well. The text should be properly changed to account for these considerations.

We searched the literature and consulted with colleagues responsible for the JPL photochemical assessments and we were unable to find any data for the rate coefficient for the reaction CO + S + M. Indeed, studies considered in the Baulch et al. (1976) review had set it equal to that for CO + O + M. However, we did find information for the reaction CS + S + M, so we have adopted the rate coefficient for CO + S + M in Krasnopolsky (2007), as it is somewhat close to the geometric mean between CO + O + M and CS + S + M.

Both eddy diffusivity profiles, ours and that of Krasnopolsky (2012) are consistent with the observations of Woo and Ishimaru (1981), i.e., $K \leq 4 \times 10^4 \text{ cm}^2 \text{ s}^{-1}$ at ~ 60 km. The profiles increasingly diverge above that level with our profile remaining reasonably consistent ($K = 1.0 \times 10^5 \text{ cm}^2 \text{ s}^{-1}$ at 86 km) with the observations of Lane and Opstbaum (1983), whereas that of Krasnopolsky (2012) is several times higher. Admittedly, there is some degree of arbitrariness in the choice of eddy diffusivity profile, which is to be expected given the sparsity of relevant data. However, our profile is meant to be consistent with theory about the relevant dynamical processes used to derive K's as described in Zhang et al (2012), p. 715. We thus see no need to alter our eddy profile.

The reviewer should note that many of the differences between models may also be due to differences in chemical schemes, the treatment of radiative transfer through an atmosphere characterized by multiple scattering and the interactions between chemistry and radiation.

In a test run, we have reduced the grid spacing between 58km and 78km to 0.2km. The comparison of the SO, SO₂ and OCS profiles between the test run and our base model is shown in the figure below. The solid lines represent the original model profiles and dashed lines represent the profiles of the test run. We can see SO, SO₂ and OCS in the test run are not significantly different from the base model. The only clear difference is the twist at around 64km in the OCS profile which is concealed by the large altitude

steps. Therefore, we reduced the grid spacing in our model and revise all the figures in our manuscript. Please see lines 81-82.

4. The previous concerns to Zhang et al. (2012) remain: (i) the nighttime $T \approx 250$ K at 95 km contradicts to temperatures of the O₂ nightglow observed by four independent teams with a mean value of 187 K at 95 km, to empirical models (VIRA, Hedin et al. 1983) that are based on the accelerometers at the entry probes, radio occultations, and sounding at 15 mu, and to VTGCM; (ii) the abundance of H₂SO₄ vapor exceeds the upper limit observed by Sandor et al. (2012) and exceeds by orders of magnitude that calculated by Krasnopolsky (2011); (iii) column photolysis rate of H₂SO₄ is close to total delivery of aerosol above 90 km.

(i) Temperatures at 94-96 km in our current model are ~ 167 K. (ii) and (iii) We have adopted constraints on the abundance of H₂SO₄ in the altitude range observed by Sandor et al. (2012). The abundance of H₂SO₄ vapor in our revised model is consistent with the upper limit observed by Sandor et al. (2012). The profiles of the important chemical species (see Figure 2 and Figure 3) are also changed due to the constraints on H₂SO₄ vapor.

Some other comments:

44: Wu et al. (2018) consider six and mention more than ten isomers.

The reviewer is correct, however, they have not yet shown to be relevant for our analysis.

49, change to: ... (Frandsen et al. 2016), and Krasnopolsky (2018) updated his model using these data. However, the predicted abundance of OSSO is smaller than that required to explain the NUV absorption of Venus by two orders of magnitude. Here we consider other possibilities of the S₂O₂ photolysis.

Done, please see lines 49-53.

53-54: the S=SO₂ dimer does not have a peak in Fig. 4 in Wu et al. This does not mean that this dimer is lacking.

Done. The reviewer is correct, but we did not explicitly rule out this possibility. To avoid misinterpretation by readers, we have simply noted that if present it was beneath detection limit in the experiments of Wu et al. Please see lines 53-59.

61: sulfur aerosol absorbs in NUV but poorly fits to the observed NUV absorption. Significant abundance of sulfur aerosol is predicted in the lower cloud layer (Krasnopolsky 2016, Icarus 274, 33-36). That paper is relevant to your problem and should be mentioned.

Reference mentioned in introduction, please see lines 39-40.

117: S + CO + M is much more effective than S + OCS in Krasnopolsky (2012, 2018).

Data are sorely needed to confirm the relative importance of these and other reactions in Venus' atmosphere. We have not been able to find experimental evidence to confirm this. Instead we have found a number of papers and reviews (e.g., Baulch et al., 1976) that take the rates as being equal.

124: r-(SO)₂ involves the production of S atoms.
Actually S₂ is removed in part by photolysis and in the reaction with O.

We have included the production of S and SO₂ by photolysis of r-(SO)₂. Loss of S₂ and by reaction with O and other species is also included in the model.

138-153: quantitative assessments of the reaction rates are lacking, and it is not clear, which reactions are either essential or negligible.

The reviewer is quite correct for many reaction pathways. We have added the following cautionary statement to remind readers that this is the case for any model. It should be remembered, though, that quantitative assessments of the rate coefficients for many reactions are lacking, to the extent that it might not be clear in many cases, whether reactions are either essential or negligible.

159: according to the Venera 14 observations at 320-390 nm, SO₂ = 10 ppm at 57 km (Krasnopolsky 1986, Photochemistry of the Atmospheres of Mars and Venus, Springer, p.152).

We note that conditions on Venus are highly variable and that care needs to be exercised in setting model input parameters, lest they correspond to some short-term and/or local perturbations to mean conditions. As indicated by the 8-year time series obtained by Venus Express, SO₂ at 70 km varies by three orders of magnitude at low latitudes. At best these data might be used as guidelines.

179-180, Figure 2: maximum values of SO₂ and SO from Sandor et al. (2010) are shown. Their mean abundances are smaller by a factor of 4, more representative, and agree with the ALMA observation (Encrenaz et al. 2015). This overestimation is used and should be corrected throughout the paper.

Done. Please see revised Figure 2.

190-192: fitting the OCS data observed near 65 km using 100 ppm at 58 km means that the model disagrees with the observations. My model for the lower atmosphere (Krasnopolsky 2013, Icarus 225, 570-580) predicts OCS = 140 ppb at 47 km, and this value is the OCS boundary condition at 47 km in Krasnopolsky (2018) that agrees with the OCS observations.

See above response to comments on line 159.

301, change to: ...that may be responsible....

Done. We thank the reviewer for catching this oversight on our part, as it more closely reflects what we meant to say. Please see line 299.

300-314: the calculated abundances of SX and SXO are rather low to explain the NUV absorption. The alternative explanation by iron chloride (Krasnopolsky 2017, Icarus 286, 134-137) should be mentioned here.

As noted above, we are not claiming that our model can explain the NUV absorption on Venus. We may also note that while FeCl₃ absorbs in the right spectral region, its absorption profile is too narrow (see Perez-Hoyos et al., JGR-Planets, 2018, 123, 145, Figure 15; reproduced above). Please see lines 314-316.

316-330: I do not share the opinion that “the sporadic nature of observations of key species precludes making definitive conclusions....”

We only wish to stress the importance of long-term measurement programs for obtaining a better understanding of the atmosphere of Venus. As shown by Venus Express, concentrations of key species like SO₂ are highly variable. This variability, which spans 3 orders of magnitude at 70 km at low latitudes, is best appreciated when examining the entire measurement period. In many cases, one-time snap shots of data elicit more questions than answers. Obviously, any short-term dataset is useful, but we should be aiming towards more, longer-term data for key species such as SO₂ and OCS. As noted above short term data, while useful, can provide only rough guidelines in this context.

The authors are the well-known authorities in Venus’ photochemistry. Maybe, some differences between our opinions on some aspects of this photochemistry are helpful for our science.

Indeed, there are differences and we thank Dr. Krasnopolsky for his willingness to be open to differing perspectives and we look forward to stimulating discussions with him on these issues. Indeed, a diversity of opinion with respect for opposing views is needed for new syntheses and progress in understanding the atmosphere of Venus.

Vladimir Krasnopolsky

REVIEWER COMMENTS

Reviewer #1 (Remarks to the Author):

The authors response to my questions are fine.

Below are my comments to the authors' responses to my initial comments.

1. Perez-Hoyos et al. (2018) apply only spectral fits of the proposed NUV absorbers and do not consider their quantitative assessments. Furthermore, they use the FeCl₃ spectrum in ethyl acetate that is very different from that in sulfuric acid (see Fig. 116 in Krasnopolsky 1986, Photochemistry of the Atmospheres of Mars and Venus, Springer). I remind that the solution of ~1% FeCl₃ in 80% H₂SO₄ was proposed to explain the NUV absorption. Therefore their conclusions are incorrect for FeCl₃ and questionable for the other species.

The model does not include the aerosol production and removal. Therefore all statements related to aerosol are hypothetical and should be properly edited (e.g. line 29 in the abstract).

The photochemical model by Krasnopolsky (2018) includes the sulfur aerosol assuming the particle radius of 0.5 μm. The calculated sulfur aerosol was tenuous, and this may be mentioned in the text.

39-40: ...in the lower and middle cloud layers...

48: it is better to change: ...the trigonal form as the lowest energy SO dimer...

2. Both the trigonal (S=SO₂) dimer and production of S₂ by photolysis of (SO)₂ are “beneath detection in the observations of Wu et al. (2018)”, and you adopt a yield of 0.1 for the latter. Therefore the reference to Wu et al. (2018) should be removed from line 66, as well as “major”.

The reactions R16 and R17 are endothermic and therefore negligible.

3. The reaction of S + CO + M is significant in the sulfur chemistry on Venus, and it is worth to add that you adopted the value of $3 \times 10^{-33} \exp(-1000/T) \text{ cm}^6 \text{ s}^{-1}$ from Krasnopolsky (2007), which is close to the geometric mean between CO + O + M and CS + S + M.

4. Reviewing the initial manuscript, I assumed that it is identical to Zhang et al. (2012) except that was written in the paper. Fortunately, some doubtful aspects of Zhang et al. (2012) have been updated. However, it is necessary to describe all these improvements. Furthermore, **a supplement with vertical profiles of the major reaction rates (similar to those in Zhang et al. 2012) would be helpful.**

Vladimir Krasnopolsky

Response to Reviewers:

We again thank both reviewers for their constructive suggestions and comments, which have significantly helped improve the manuscript.

We have taken into account all the comments and revised the paper accordingly. In the following, we reply to the comments point by point. Our responses are highlighted.

Below are my comments to the authors' responses to my initial comments.

1. Perez-Hoyos et al. (2018) apply only spectral fits of the proposed NUV absorbers and do not consider their quantitative assessments. Furthermore, they use the FeCl₃ spectrum in ethyl acetate that is very different from that in sulfuric acid (see Fig. 116 in Krasnopolsky 1986, Photochemistry of the Atmospheres of Mars and Venus, Springer). I remind that the solution of ~1% FeCl₃ in 80% H₂SO₄ was proposed to explain the NUV absorption. Therefore their conclusions are incorrect for FeCl₃ and questionable for the other species.

The model does not include the aerosol production and removal. Therefore all statements related to aerosol are hypothetical and should be properly edited (e.g. line 29 in the abstract).

The photochemical model by Krasnopolsky (2018) includes the sulfur aerosol assuming the particle radius of 0.5 μm. The calculated sulfur aerosol was tenuous, and this may be mentioned in the text.

We have not claimed in our manuscript that we have identified the UV absorber. We have only said that FeCl₃, SnO and polysulfur are possible candidates that may be responsible for the UV absorber in Venus' atmosphere. We cited Perez-Hoyos et al 2018 as an exemplary reference for further information on the possible candidates. The reviewer's concerns about the conclusions reached by Perez-Hoyos et al 2018 may be appropriate for an article by the reviewer but are not relevant to our manuscript.

For the aerosol related statements, please see lines 28 and 317-319. We note that we expect a sizable fraction of these species to be in the aerosol phase based on their thermochemical properties.

We mention Krasnopolsky (2018) in lines 38-40.

39-40: ...in the lower and middle cloud layers...

Done. See lines 38-40.

48: it is better to change: ...the trigonal form as the lowest energy SO dimer...

Done. See lines 47-48.

2. Both the trigonal (S=SO₂) dimer and production of S₂ by photolysis of (SO)₂ are “beneath detection in the observations of Wu et al. (2018)”, and you adopt a yield of 0.1 for the latter. Therefore the reference to Wu et al. (2018) should be removed from line 66, as well as “major”. Here we said that the trigonal form of the dimer and S₂ weren't observed to be primary products. This does not necessarily rule out its hypothetical production in a small yield as we noted. Wu et al. (2018) did state that S=SO₂ might be involved in the Venusian atmosphere. Here we remove ‘major’ and keep the Wu et al. (2018) reference.

The reactions R16 and R17 are endothermic and therefore negligible.

We have removed R16 and R17 in our model. The figures are adjusted accordingly.

3. The reaction of $S + CO + M$ is significant in the sulfur chemistry on Venus, and it is worth to add that you adopted the value of $3 \times 10^{-33} \exp(-1000/T) \text{ cm}^6 \text{ s}^{-1}$ from Krasnopolsky (2007), which

is close to the geometric mean between $CO + O + M$ and $CS + S + M$.

Please see line 91-93.

4. Reviewing the initial manuscript, I assumed that it is identical to Zhang et al. (2012) except that was written in the paper. Fortunately, some doubtful aspects of Zhang et al. (2012) have been updated. However, it is necessary to describe all these improvements. Furthermore, a supplement with vertical profiles of the major reaction rates (similar to those in Zhang et al. 2012) would be helpful.

Thanks for the suggestions. Please see lines 87-90, 171-172 and Figure S1.

Vladimir Krasnopolsky

REVIEWERS' COMMENTS

Reviewer #2 (Remarks to the Author):

The authors have adequately responded to my latest comments, and I support publication of the manuscript.

Vladimir Krasnopolsky